# Estimation of the Actual Incidence of Coronavirus Disease (COVID-19) in Emergent Hotspots: The Example of Hokkaido, Japan during February–March 2020

**DOI:** 10.3390/jcm10112392

**Published:** 2021-05-28

**Authors:** Andrei R. Akhmetzhanov, Kenji Mizumoto, Sung-Mok Jung, Natalie M. Linton, Ryosuke Omori, Hiroshi Nishiura

**Affiliations:** 1Graduate School of Medicine, Hokkaido University, Kita 15 Jo Nishi 7 Chome, Kita-ku, Sapporo-shi, Hokkaido 060-8638, Japan; akhmetzhanov@gmail.com (A.R.A.); seductmd@gmail.com (S.-M.J.); nlinton@gmail.com (N.M.L.); 2Global Health Program, Institute of Epidemiology and Preventive Medicine, College of Public Health, National Taiwan University, 17 Xu-Zhou Road, Taipei 10055, Taiwan; 3Graduate School of Advanced Integrated Studies in Human Survivability, Kyoto University, Yoshida-Nakaadachi-cho, Sakyo-ku, Kyoto 606-8501, Japan; mizumotokenji@gmail.com; 4Hakubi Center for Advanced Research, Kyoto University, Yoshidahonmachi, Sakyo-ku, Kyoto 606-8306, Japan; 5School of Public Health, Kyoto University, Yoshidakonoe cho, Sakyo ku, Kyoto 606-8501, Japan; 6Research Center for Zoonosis Control, Hokkaido University, Kita 19 Jo Nishi 10 Chome, Kita-ku, Sapporo-shi, Hokkaido 001-0019, Japan; r.omori12@gmail.com; 7Core Research for Evolutional Science and Technology (CREST), Japan Science and Technology Agency, Honcho 4-1-8, Kawaguchi, Saitama 332-0012, Japan

**Keywords:** epidemiology, travel medicine, COVID-19, emerging infectious diseases

## Abstract

Following the first report of the coronavirus disease 2019 (COVID-19) in Sapporo city, Hokkaido Prefecture, Japan, on 14 February 2020, a surge of cases was observed in Hokkaido during February and March. As of 6 March, 90 cases were diagnosed in Hokkaido. Unfortunately, many infected persons may not have been recognized due to having mild or no symptoms during the initial months of the outbreak. We therefore aimed to predict the actual number of COVID-19 cases in (i) Hokkaido Prefecture and (ii) Sapporo city using data on cases diagnosed outside these areas. Two statistical frameworks involving a balance equation and an extrapolated linear regression model with a negative binomial link were used for deriving both estimates, respectively. The estimated cumulative incidence in Hokkaido as of 27 February was 2,297 cases (95% confidence interval (CI): 382–7091) based on data on travelers outbound from Hokkaido. The cumulative incidence in Sapporo city as of 28 February was estimated at 2233 cases (95% CI: 0–4893) based on the count of confirmed cases within Hokkaido. Both approaches resulted in similar estimates, indicating a higher incidence of infections in Hokkaido than were detected by the surveillance system. This quantification of the gap between detected and estimated cases helped to inform the public health response at the beginning of the pandemic and provided insight into the possible scope of undetected transmission for future assessments.

## 1. Introduction

In December 2019, a cluster of 41 patients with atypical pneumonia of unknown etiology was reported in the city of Wuhan, China [1,2]. The number of atypical pneumonia cases in Wuhan rapidly increased in early January and cases began appearing across China and in other countries [3,4]. The cause of the pneumonia was recognized as a newly emerged coronavirus called severe acute respiratory syndrome coronavirus 2 (SARS-CoV-2), and the disease it causes was given the name coronavirus disease 2019 (COVID-19). Cases of COVID-19 have now been reported in more than 120 countries worldwide [5], and the WHO declared it a pandemic on 11 March 2020. By the end of 2020, the disease had spread to nearly every country in the world. In the present paper, we examine the emergence of a COVID-19 hotspot in Hokkaido Prefecture, Japan, during the first wave of the pandemic. Because of the limited testing capacity in the first months of the outbreak [6] and the specific contact-based, centered surveillance of SARS-CoV-2 infections in Japan [7], a substantial fraction of infected individuals may have remained undetected during the first months of the pandemic.

Control of COVID-19 spread is complicated by various factors, including nonspecific clinical symptoms, especially during the early phase of infection [8,9,10]; a relatively high proportion of pre-symptomatic and asymptomatic infections [11,12,13,14,15]; multiple possible routes of transmission—including aerosol, direct contact, and fecal–oral transmission [10,16,17]—as well as the moderate to high potential of superspreading events generated by a small proportion of cases [18,19]. Additionally, the incubation period can vary widely [20,21], and re-occurrence of the disease after primary infection is possible [22]. Nonetheless, the transmission dynamics of COVID-19 in the community can be roughly grasped by exploring epidemiological indicators such as the number of confirmed infections in a given location, or the number of exported cases.

Clinical manifestation of COVID-19 begins with nonspecific symptoms (similar to seasonal influenza), which remain mild for most persons. Some infections may remain entirely asymptomatic. For a small fraction of symptomatic cases, the disease may resolve into a life-threatening onset of acute respiratory distress [19,23]. Despite efforts towards mitigation, the number of confirmed cases worldwide represents only a fraction of all infections, and the actual number of cases including mild and asymptomatic infections may be as large as four- or ten-fold [15,24,25]. Hence, the derivation of reliable estimates of the actual number of cases is needed to conduct risk assessments of the spread of COVID-19 and inform policies related to containment and mitigation [26]. This was especially important for newly emerging hotspots in Japan and some other countries at the beginning of 2020 [4,27].

The increased number of cases in and exportation of cases from Hokkaido in February–March 2020 signaled an urgent need to increase containment efforts within the prefecture. To help guide the policymaking process and better understand the risk of further spread within and exportation from Hokkaido, we derived real-time estimates of the actual incidence by accounting for under-ascertained cases. The methods can be further expended to other areas or countries experiencing a surge of new cases of COVID-19, any other emergent disease, or their new variants.

## 2. Methods

### 2.1. Epidemiological Data

Three sources of data were leveraged to estimate the actual incidence of COVID-19 in Hokkaido. Firstly, cases exported domestically and internationally from Hokkaido were identified based on government reports. The domestically exported cases were Kumamoto (1 case), Nagano (1 case), and Chiba (1 case) prefecture residents [28,29,30]. The internationally exported cases were reported from Malaysia (1 case) and Thailand (2 cases) [31,32]. Considering that the two Thailand cases were husband and wife and one case was asymptomatic, one infection may have been the result of household transmission, as similarly described in [33], rather than exported from Japan. Secondly, the total volume of air passengers from all airports in Hokkaido to Thailand and Malaysia over the two-month period of January and February 2019 was retrieved from the International Air Transport Association (IATA) and included 9349 passengers flown directly to Malaysia and 45,137 to Thailand. Lastly, the population size of Hokkaido was obtained from the Hokkaido Prefectural Government website [34].

The calculation of COVID-19 incidence in Sapporo was based on cases of non-Sapporo residents who (i) became infected in Sapporo and were diagnosed outside of Sapporo or (ii) cases diagnosed outside of Sapporo who had no information on their source of infection (i.e., unknown link). Cases who were infected outside of Sapporo by Sapporo residents were excluded from the analysis, as they did not represent true exportation of infection from Sapporo by a local case. For example, in one instance, a Sapporo resident travelled to Kitami city in Okhotsk subprefecture and caused secondary infections among Kitami residents. These Kitami residents were excluded from our analysis, as they were infected outside of Sapporo. However, if a case from Kitami were to travel to Sapporo, have likely been infected there, and then be subsequently diagnosed in Kitami, they would be eligible for our analysis. Data on cases reported through the end of February 2020, including dates of illness onset, subprefecture of diagnosis, history of travel to Sapporo, and epidemiologic linkage to other confirmed cases, were retrieved from the websites of government entities in Hokkaido.

The estimate of the fraction of commuters between subprefectures was obtained with reference to a survey of the daytime and nighttime population sizes in Sapporo [35]. By subtracting the daytime population size from the nighttime population size and dividing the difference by the nighttime population, we obtained a value of 3.7% [35], reflecting the change in the population size of Sapporo due to commuting. However, this estimate accounts for the movement of individuals both within Ishikari subprefecture (including Sapporo and several other cities) and between subprefectures. In contrast, our model requires the average estimate between all subprefectures of Hokkaido. Therefore, we set this parameter at 1% and performed a sensitivity analysis wherein we varied the fraction of commuters between 0.5% and 4%.

### 2.2. Modeling Commuter Movement between Subprefectures of Hokkaido

Two single-parameter models of human movement were employed to predict the rate of commuting between Sapporo and the various subprefectures of Hokkaido—a radiation model and a model of uniform selection. Both emphasize the attractiveness of large population centers [36,37,38,39]. The uniform selection model [40] predicts the commuting volume using the following formula:(1)$$M_{ij}=M_{i}\frac{P_{i}}{N-Q_{j}}\;,$$
where $P_{i}$ is the population size at the origin i, $Q_{j}$ is the population size at the destination j, and N is the total population of Hokkaido. The only parameter to be estimated is $M_{i}$, which defines the scaling of the ongoing number of commuters from the source subprefecture and is equal to the proportion of commuters multiplied on the population size $P_{i}$ [39]. In contrast, the radiation model [41] incorporates the distance metric between the origin and the destination and defines a commuting volume with the following formula:(2)$$M_{ij}=M_{i}\frac{P_{i}Q_{j}}{\left(P_{i}+R_{ij}\right)\left(P_{i}+Q_{j}+R_{ij}\right)}$$
where $R_{ij}$ denotes the total population within a radius around two centers, $P_{i}$ and $Q_{j}$. Both matrices $M_{ij}$ (1)–(2) were made symmetric in our simulations. Either model (1) or (2) was selected based on the values of a widely applicable information criterium (WAIC), with a lower value regarded as better.

An aggregated dataset of population sizes and central point coordinates for each subprefecture was used to estimate the commuting rates using the R package “movement” [38]. The package required a fraction of the commuting individuals to be a single input parameter for the function. Due to the uncertainty of this value, we performed a sensitivity analysis with reference to the daytime and nighttime populations of Sapporo [35].

### 2.3. Estimating Actual Incidence of COVID-19 Cases in Hokkaido Prefecture and Sapporo City

The actual incidence was estimated in (i) Hokkaido Prefecture using the number of detected cases among international and domestic travelers outbound from Hokkaido, and in (ii) Sapporo city using the number of confirmed cases within different subprefectures of Hokkaido. Both estimates were thought to be close to each other, because Sapporo was the epicenter for COVID-19 spread within Hokkaido.

The framework was based on two key assumptions. First, the outbreak was considered to be in its initial stage during the time period of February to March 2020, when the first cases in Sapporo emerged and had just expanded to other areas in Hokkaido. In such an instance, the disentanglement of locally acquired infections from the imported cases was still possible in each subprefecture. Second, the travel volume between subprefectures in Hokkaido was assumed to be well-approximated by the models of commuting movement, which were described above.

(i)Estimating incidence in Hokkaido by using imported cases among travelers outbound from Hokkaido

A balance equation was implemented to predict incidence across all Hokkaido using the case count of air passengers from Hokkaido with international or domestic destinations [3,42,43,44]. Given the observed cumulative count of exported cases C, we estimated the fraction of infected individuals in Hokkaido $\hat{p}$ by the following equation:(3)$$\hat{p}=C\frac{365}{mT}\;,$$
where *m* is the total volume of passengers to the corresponding destination and *T* is the infectious period, approximated from the observed average virus shedding period of 5.0 days [45]. Accordingly, the incidence in Hokkaido was defined by $\hat{p}n$, where n is the catchment population size for the international airport in Hokkaido. To account for stochasticity, we used a binomial sampling process, maximum likelihood estimation, and 95% confidence intervals derived from the profile likelihood.

(ii)Estimating incidence in Sapporo using cases imported within Hokkaido

Following a previously developed framework [46], the number of infections $C_{j}$ in subprefecture j of Hokkaido Prefecture was sampled from a negative binomial distribution with the mean $\lambda_{j}$ linearly proportional to the predicted travel volume between Sapporo and subprefecture j: $\lambda_{j}=\beta \cdot M_{\circ j}$, and the dispersion parameter k:(4)$$C_{j}\;\sim \;\mathrm{Negative}\;\mathrm{Binomial}\;\left(\mathrm{mean}=\lambda_{j},\mathrm{overdispersion}=k\right),\;\lambda_{j}=\beta \cdot M_{\circ j},\;\;j=1\ldots 14,$$
where the regression coefficient β is considered to be independent of a subprefecture and to be fitted to the data.

Because Sapporo is a part of Ishikari subprefecture, for our study, we defined Ishikari subprefecture as all cities within Ishikari subprefecture, excluding Sapporo (Figure 1B). The flow $M_{\circ j}$ was predicted using the uniform selection and radiation models of human movement [40,41], as described in Equations (1) and (2). The cumulative incidence in Sapporo was then estimated by extending Equation (4) to the entire population of Sapporo:(5)$$C_{\circ }\;\sim \;\mathrm{NegativeBinomial}\left(\beta \cdot N_{\circ },\;k\right),$$
where $N_{\circ }$ is the population size of Sapporo.

The derived estimates of the linear regression model (3) were also used to validate our assumption of linearity between the number of reported COVID-19 infections in each subprefecture and the daily commuting rate [47]. Using the fit of the predicted travel volume and observed case counts, the best-fit model between (1) and (2) was selected based on the difference in the WAIC values.

### 2.4. Reporting Delay between Illness Onset and Confirmation

Because the dataset for this study was collected at a near-real-time setting, the assessment of the under-ascertainment of reported cases was made by evaluating the reporting delay distribution. For this purpose, the time interval from illness onset day $S_{i}$ to report of case confirmation day $R_{i}$ for each confirmed case i was extracted from the data. The doubly interval-censored likelihood function ℒ [20,47] was implemented for estimation:(6)$$ℒ(\theta \;|\;D)=\underset{i}{\prod }\int_{S_{i}}^{S_{i}+1}\int_{R_{i}}^{R_{i}+1}h\left(s\right)\cdot f\left(r-s\;|\;\theta \right)\;dr\;ds.$$

Here, h(.) is the probability distribution function (PDF) of illness onset time following a uniform distribution, and f(. | θ) is the PDF of the reporting delay independent of h(.). D represents a dataset among all confirmed cases i, where both the time of illness onset s and the time of case confirmation r are defined with precision to one day, i.e., they belong to the time intervals $\left(S_{i},S_{i}+1\right)$ and $\left(R_{i},R_{i}+1\right)$, respectively. The distribution to be defined is f(. | θ), and it was fitted to gamma, lognormal, and Weibull distributions, each with a set of parameters θ. The best-fit distribution was then selected by comparing the WAIC values and selecting the one with a lower value.

We note that an alternative way to implement a right-truncated likelihood is by using the following formula:(7)$$ℒ(\theta \;|D)=\underset{i}{\prod }\int_{S_{i}}^{S_{i}+1}\int_{R_{i}}^{R_{i}+1}h\left(s\right)\cdot \frac{f\left(r-s\;|\;\theta \right)}{F(t^{*}-s\;|\;\theta )}\;dr\;ds,$$
where $t^{*}$ is the cut-off time at noon on 29 February 2020, and F(. | θ) is the cumulative distribution function of f(. | θ). However, our analysis has shown that the mean delay was longer than two weeks, and this was inconsistent with real observations. The reason is that the right truncation does not account for the effect of control measures implemented in late February (see Discussion).

### 2.5. Simulation Platform

The data were processed using R version 3.6.2 and Python version 3.6.10. Markov chain Monte Carlo (MCMC) simulations were performed in Stan (cmdStan version 2.22.1 [48]) for estimation of the delay distribution, and in PyMC3 version 3.8 [49] for all other estimates. The code is available online [50].

## 3. Results

### 3.1. Epidemiological Situation

The first case was reported in Hokkaido on 28 January 2020. One month later, on 28 February 2020, the total count of confirmed cases reached 65, with 13 cases reported by Sapporo and another 51 cases reported by 11 of the 14 subprefectures of Hokkaido (and one elsewhere). The place of diagnosis for the first case was unspecified. Initially, cases were predominantly linked to the Sapporo Snow Festival, but overall, the geographic distribution of COVID-19 cases was widespread. By 28 February, Japan had also been notified of three cases diagnosed in Thailand and Malaysia who were believed to have been infected in Hokkaido, as well as three domestic cases likely exposed in Hokkaido that were reported by other prefectures (Table 1, Figure 1A).

A surge of cases was observed at the end of February (Figure 2A). In the first half of February, peaks of reported cases by date of illness onset were separated by 3–4 days on average, consistent with estimates of the serial interval [51]. The epidemic curve peaked around 18 February 2020, but the subsequent decline in cases can be explained by the delay in reporting. The delay distribution fitted with the gamma distribution had a mean of 7.9 days (95% CI: 6.9–9.0) and a standard deviation of 4.2 days (95% CI: 3.3–5.2). The 95th percentile was 15.6 days (95% CI: 13.4–18.8), implying that cases with illness onset in the last two weeks of February were likely to be under-ascertained.

### 3.2. Estimated Incidence in Hokkaido Using Confirmed Cases Diagnosed Outside Hokkaido (Method (i))

The first three cases diagnosed in early February outside of Hokkaido indicated a low incidence of COVID-19 in Hokkaido with an estimated upper bound (95th percentile) of fewer than 100 cases. Our estimate of the cumulative incidence in Hokkaido as of 25 February 2020 is dependent on whether the transmission within the infected couple who travelled to Thailand was acquired in the community or within the household, and we estimated the incidence in Hokkaido to be higher in the former scenario, at 3446 cases (95% CI: 857–8931), compared to the latter, with 2297 cases (95% CI: 382–7091); see Figure 3A.

### 3.3. Estimated Incidence in Sapporo Using Confirmed Cases Diagnosed within Hokkaido (Method (ii))

A total of 34 Hokkaido cases were included in our model (Figure 1B). We fitted the radiation and uniform selection models to the population data for each subprefecture based on their centroids (Appendix A). The resulting fit to the observed incidence of COVID-19 cases showed that the uniform selection model performed better than the radiation model (WAIC values: 60.2 vs. 68.3, respectively). Figure 3B shows the fit using the model with uniform selection. If we assume the fraction of commuters to be at 1%, then the mean estimated incidence in Sapporo was 2233 cases (95% CI: 0–4893) as of 28 February 2020. Variation in the fraction of commuters between 0.5% and 4% results in an estimated incidence between 4440 (95% CI: 0–9687) and 563 (95% CI: 0–1221) for 0.5% and 4%, respectively (Figure 3C).

## 4. Discussion

Our estimate of the incidence in Sapporo in February–March 2020 is in the range of 1000–10,000 cases and resembles early estimates of COVID-19 incidence in Wuhan city, China that used data on the first cases among international travelers [3,52]. Assuming, however, that our estimates are correct, a substantial proportion of these infected persons likely had only a mild or asymptomatic course of the disease [11,21], did not seek medical care, and were not detected by the surveillance system. The median estimated infection-to-confirmed-case ratio in Hokkaido was 26:1 (95% CI: 4.2–78.8), which includes, but is expectedly higher than, the 7:1 (95% CI: 5.5–10.0) estimate for the first wave of infections in the United States [25]. However, in the early stage of the pandemic, the case-finding strategy in Japan was based on extensive contact tracing [7], and differences in adopted case definitions between the two countries may have contributed to the observed difference in estimates. Our inference method was also less specific and did not rely on a compartmental Susceptible–Infected–Recovered (SIR) model as in [25].

Use of data on the travel volume of commuters and tourists within Hokkaido could be beneficial for fitting the travel volume to the observed incidence rather than adopting a model of human movement (Figure 3B). However, publicly available data on train, bus, and private car travel in Hokkaido were inconsistent and scarce, and we were unable to use them for this study. Nevertheless, our results demonstrate that the use of mathematical models of human movement at the scale of estimating the travel volume between administrative subunits of a given region is a promising alternative to the use of observed transportation data. Previously, this was also demonstrated by other researchers in assessing the risk of spread of yellow fever in Angola [36] or during epidemics in resource-poor settings [37].

In addition, surveillance among travelers at international borders may detect infections not found through standard surveillance systems due to the implementation of additional detection methods [53,54]. As shown by other researchers [46,55], the linear regression fit of the travel volume among international travelers to observed disease incidence represents an efficient way to distinguish areas that are able to detect most new infections from those that are likely to miss a larger fraction of cases. One limitation of our approach is that we were unable to distinguish international travelers from Hokkaido to Malaysia and Thailand who actually stayed in those countries from those who transited to another international destination. We applied the same approach to compare the performance of the various subprefectures of Hokkaido, as shown in Figure 3B. Subprefectures that fall within the 95% CI of the linear regression line are considered to have surveillance systems that succeed in detecting COVID-19 infections within their jurisdiction. Subprefectures that fall below the 95% CI of the linear regression line may not have surveillance systems that are sensitive enough to detect all imported cases [46]. The four subprefectures with no reported cases by the end of March were Rumoi, Soya, Nemuro, and Shiribeshi. The first three are distant and less connected to Sapporo compared to, for example, Kamikawa or Oshima, which contain the second and third largest cities of Hokkaido, Asahikawa and Hakodate (Figure 1B). However, Shiribeshi is located next to Ishikari subprefecture/Sapporo and includes the relatively large city of Otaru. The lack of reports in Shiribeshi may be explained by our use of place of diagnosis over place of residence for determining the subprefecture assigned to cases. This was done because the specific place of residence was not always reported. We suspect that some cases residing in Shiribeshi subprefecture were diagnosed in Ishikari subprefecture.

Our analysis of the reporting delay did not implement right truncation of the likelihood because the low case count (by date of illness onset) seen in late February through the beginning of March 2020 was highly likely due to the effect of control measures in Hokkaido rather than delays in reporting. In support of this assumption, there were fewer cases reported between 28 February and 6 March 2020 compared to the week of 21–27 February 2020 (cf. Figure 2 and Appendix A). Right truncation should, therefore, be used with caution when it is unclear whether recent incidence is increasing or decreasing.

In conclusion, our study demonstrates that combining data sources of imported COVID-19 cases at different (sub-prefectural, inter-prefectural, or international) levels represents a valuable way for estimating the actual incidence at the origin. Based on this approach, we showed that the under-ascertainment rate of COVID-19 cases in Hokkaido Prefecture at the beginning of the pandemic in February–March 2020 was much higher than the analogous estimate from the United States in the corresponding period [25]. This could be explained by the different case-finding strategy used in Japan in early 2020, when mass testing of the public was rarely implemented and only the contacts of confirmed cases were investigated by the authorities [6,7]. Preliminary analysis of the offspring distribution among infected individuals in the detected clusters around Japan has shown that approximately 80% of cases do not produce secondary infections (i.e., their reproduction number is equal to zero) [56]. Because most of the clusters have been linked to the known cases and could be traced back in time, we argue that there is still a window of opportunity for containment of the disease, and only some small fraction of the infections may drive the epidemic [18]. We agree with other studies, e.g., [57,58], that a successful strategy for the control of COVID-19 lies in strict movement restriction, avoidance of social gatherings, and an intense investigation effort in contact tracing with the possible help of new information technologies such as digital assistance for contact tracing [59,60].

## Figures and Tables

**Figure 1 jcm-10-02392-f001:**
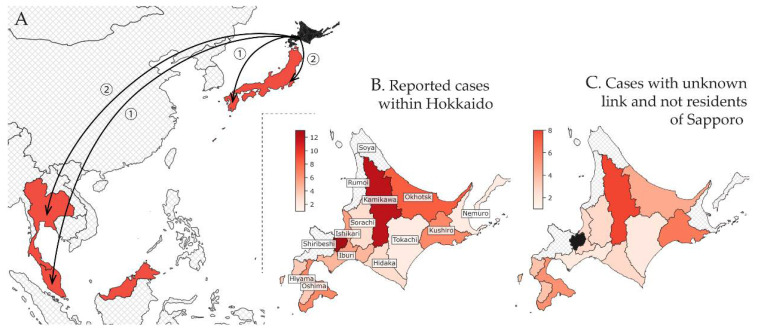
Geographical distribution of confirmed cases linked to Hokkaido by subprefecture of diagnosis as of 29 February 2020. (**A**) International and domestic cases (excluding Hokkaido). Hokkaido is shaded in black, whereas affected countries are depicted in red. The circled numbers indicate the total count of cases for each import destination. (**B**) All cases confirmed in Hokkaido. The labels indicate the names of the subprefectures of Hokkaido. (**C**) Only cases included in our analysis: non-residents of Sapporo with unknown links either with or without history of travel to Sapporo. Ishikari subprefecture was separated into two subregions: Sapporo city and outside Sapporo city. Hatched areas in grey indicate subprefectures with zero counts.

**Figure 2 jcm-10-02392-f002:**
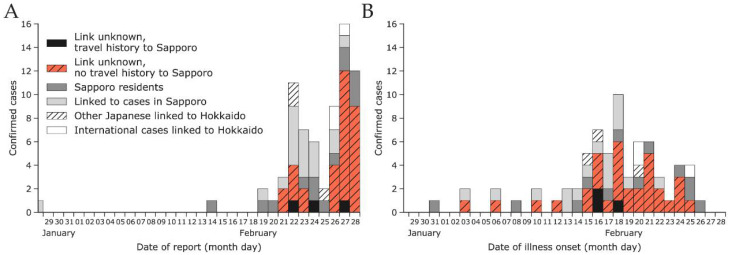
Epidemic curves by date of report (**A**) and date of illness onset (**B**) as of 29 February 2020 for confirmed cases among Japanese nationals that were linked to Hokkaido. The total number of cases is characterized according to the legend shown in (**A**).

**Figure 3 jcm-10-02392-f003:**
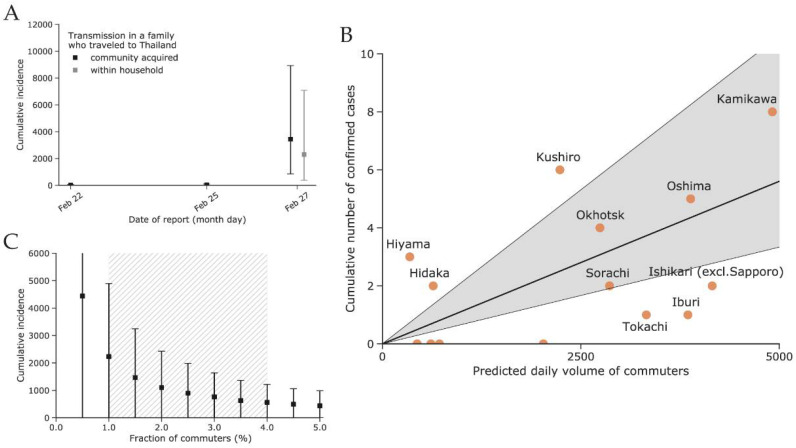
(**A**) Estimated cumulative incidence of coronavirus disease 2019 (COVID-2019) in Hokkaido using the count of imported cases either internationally or domestically outbound from Hokkaido (method (i)). (**B**) Modeled daily volume of commuters and observed incidence of COVID-19 by subprefecture of diagnosis. The four subprefectures with zero case counts are Rumoi, Soya, Nemuro, and Shiribeshi (from left to right). (**C**) Estimated cumulative incidence in Sapporo as of 29 February with a varying fraction of commuters (method (ii)). The plausible range between 0.5% and 4% is indicated by the hatched grey area. The square points in (**A**,**C**) indicate the estimated mean, whereas the error bars indicate the 95% confidence intervals.

**Table 1 jcm-10-02392-t001:** Exportation events and estimated incidence in Hokkaido by date of report.

Case Number	Diagnosis Location	Date of Illness Onset	Date of Report	Cumulative Count	Estimated Incidence in Hokkaido (95% CI)
1	Kumamoto	15 February 2020	22 February 2020	2	24 (4–74)
2	Chiba	16 February 2020			
3	Nagano	20 February 2020	25 February 2020	3	36 (9–93)
4–5	Thailand	20 February 2020	26 February 2020	6	3446 (857–8931) ^#^ 2298 (382–7091) ^##^
		NA			
6	Malaysia	25 February 2020	27 February 2020		

CI, confidence interval (the 95% CI was derived from profile likelihood); NA, not available. The estimated incidence is updated by the function of the date of report. The estimated incidence for the latest available reporting date (27 February 2020) depends on whether both cases 4 and 5 acquired their infection in Hokkaido (^#^) or whether one was infected by the other (^##^).

## Data Availability

Publicly available datasets were analized in this study. This data can be found here [50].

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
