# Peer review of "Estimation of the Actual Incidence of Coronavirus Disease (COVID-19) in Emergent Hotspots: The Example of Hokkaido, Japan during February–March 2020"

_jcm, 2021, doi:10.3390/jcm10112392_

Round 1

Reviewer 1 Report

Accepted.

The authors have addressed the reviewer's concerns and observations, stated in the first round of reviews.

Author Response

Accepted. The authors have addressed the reviewer's concerns and observations, stated in the first round of reviews.
>> We thanks the reviewer for their time in assessment of our manuscript.

Reviewer 2 Report

The paper proposes  to predict the actual number of COVID-19 cases in (i) Hokkaido Prefecture and (ii) Sap- 26poro City using data on cases diagnosed outside these areas.  My comments are:

1.     I would welcome a much broader discussion about the methods and approaches chosen.

2.     Simulation platform is not explained.

3.     They give a lot of terminology and equations without explaining which term is what in Page 3.

4.     The main motivation about the COVID model will need to explain.

5.     The conclusion will need to rewrite with emphasize contributions and results.

6.     Each simulation Fig will need to clearly explain.

7.     The mathematical term used in proposed prediction method need to be clearly explain 8. The system level block diagram in the introduction section can be included

Author Response

The paper proposes to predict the actual number of COVID-19 cases in (i) Hokkaido Prefecture and (ii) Sapporo City using data on cases diagnosed outside these areas.
>> We thank the reviewer for their time assessing our manuscript and providing helpful comments.

1, 7, 8).  I would welcome a much broader discussion about the methods and approaches chosen. The mathematical term used in proposed prediction method need to be clearly explain. The system level block diagram in the introduction section can be included.
>> We apology for some confusion in presenting our methodology. To better address the raised issues by the reviewer, we re-structured the Methods section and make clearer the protocol for our conducted analysis. Now the Results section has the same story flow as the Methods, so it is more convenient to follow all steps.

2). Simulation platform is not explained.
>> Although the paragraph outlining the used software/simulation platform has been present, we specified it with a new subsection “Simulation platform” in the Methods section.

3). They give a lot of terminology and equations without explaining which term is what in Page 3.
>> We apology for being brief, we added more mathematical details to our methods. Please see a new information under the first formula in Section 2.4.

4). The main motivation about the COVID model will need to explain.
>> We modified the last sentence in the abstract to better outline our main motivation of the study: “This quantification of the gap between detected and estimated cases helped to inform public health response at the beginning of pandemic and provided insight into the possible scope of undetected transmission for future assessments.” Additionally, we repeat the same statement in the Introduction (see two last paragraphs for our motivation).

5). The conclusion will need to rewrite with emphasize contributions and results.
>> We added a new concluding paragraph at the end of our discussion, whereas the very last paragraph outlines the importance of the findings. One of the previous ending paragraphs was moved to the beginning of the Discussion for better clarity of the structure.

6. Each simulation Fig will need to clearly explain.
>> We specified the applied method in the caption to Figure 3 (method (i) or method (ii)).

Reviewer 3 Report

The authors report data from models aimed to predict the actual number of COVID-19 cases in Hokkaido Prefecture and Saporo City using data on cases diagnosed outside these areas. The manuscript is well written and interesting. I have the following comments:

1) ABSTRACT:  I think that a bit more details on how these estimates were carried out should be reported

2) Line from 74 to 90: I suggest to mode these lines to an overall paragraph in the methods section

3) I think that the real merit of the methods of these models can be appreciated by an epidemiologists with experience in pandemic modelling

4) Discussion: I suggest the authors to insert one or two sentence that sum, practically, the meaning of your findings and its importance.

5) A conclusion paragraph should be added

6) Funding: The authors reported funding from German Federal Ministry of Health (BMG) COVID-19 Research and Development but all the authors are from Japan or Twain. Could they comment on this?

Author Response

The authors report data from models aimed to predict the actual number of COVID-19 cases in Hokkaido Prefecture and Sapporo City using data on cases diagnosed outside these areas. The manuscript is well written and interesting.
>> We thank the reviewer for their time assessing our manuscript and giving a positive evaluation.

1) ABSTRACT:  I think that a bit more details on how these estimates were carried out should be reported
>> We added the required sentence to our abstract.

2) Line from 74 to 90: I suggest to move these lines to an overall paragraph in the methods section
>> We agree with the reviewer: the moved paragraph indeed fits better the methods section.

3) I think that the real merit of the methods of these models can be appreciated by epidemiologists with experience in pandemic modelling
>> We added a concluding remark to our Introduction stating that “our methods can be further expanded to other areas and countries that experience a surge of new cases for an emergent disease or its new variants.”

4-5) Discussion: I suggest the authors to insert one or two sentence that sum, practically, the meaning of your findings and its importance. A conclusion paragraph should be added
>> We added a concluding paragraph at the end with the last paragraph outlining the importance of our findings. We also moved one of the previous ending paragraphs to the beginning of the Discussion to make the structure clearer.

6) Funding: The authors reported funding from German Federal Ministry of Health (BMG) COVID-19 Research and Development but all the authors are from Japan or Twain. Could they comment on this?
>> We were happy to receive some financial support from WHO linked to German Federal Ministry of Health. This was reflected in our Acknowledgments section accordingly.